# Two Modified Questionnaires for the Assessment of Nutrition Impact Symptoms in the Rehabilitation Phase after Burn Injury: A Content Validation Study

Josefin Dimander [1,2,*], Agneta Andersson [3], Adriana Miclescu [4] and Fredrik Huss [1,2]

1   Department of Surgical Sciences, Plastic Surgery, Uppsala University, 751 85 Uppsala, Sweden;
    fredrik.huss@akademiska.se
2   Burn Centre, Department of Plastic and Maxillofacial Surgery, Uppsala University Hospital,
    751 85 Uppsala, Sweden
3   Department of Food Studies, Nutrition and Dietetics, Uppsala University, 751 22 Uppsala, Sweden;
    agneta.andersson@ikv.uu.se
4   Multidisciplinary Pain Centre, Department of Surgical Sciences, Uppsala University, 751 85 Uppsala, Sweden;
    adriana.ana.miclescu@surgsci.uu.se
*   Correspondence: josefin.dimander@akademiska.se; Tel.: +46-(0)-736-311-930

**Abstract:** Disease Related Appetite Questionnaire (DRAQ) and Eating Symptom Questionnaire (ESQ) are used to assess nutrition impact symptoms, which are symptoms that can negatively affect the patients' food intake. However, these questionnaires have not yet been adapted to the needs of patients recovering from burn injuries. Our aim was therefore to develop DRAQ and ESQ for assessments of nutrition impact symptoms after burn injury. A content validation index (I-CVI) for items included in DRAQ and ESQ, regarding their relevance for possible nutrition impact symptoms in a burn-injured patient (Likert scale 1–4), was performed by an expert review group. A clarity validation by expert and non-expert reviewers was carried out. Two of the eleven questions in DRAQ and eight of the fourteen questions in ESQ were not considered relevant and were therefore removed from the questionnaires. Five additional questions were added to DRAQ and two to ESQ. A high degree of consensus on relevance (scale-content validity index average, S-CVI/Ave, 0.86 for DRAQ-burn and 0.83 for ESQ-burn) was reached in the expert group. To conclude, it is suggested that we use developed forms of DRAQ and ESQ (DRAQ-burn and ESQ-burn) for the assessment of nutrition impact symptoms, specifically during the rehabilitation phase of burn-injured patients.

**Keywords:** burn injury; surveys and questionnaires; questionnaire validation; nutrition impact symptoms; DRAQ; ESQ

## 1. Introduction

A burn injury is a complex trauma followed by the body's increased demand for energy and protein due to the hypermetabolic response, which can persist from months to years after injury [1–3]. An adequate nutritional intake is important to optimise wound healing as well as to prevent muscle wasting [4]. A better understanding of the underlying mechanisms associated with nutritional intake is of importance to the outcome after such an injury.

A broad variety of symptoms can negatively affect the patients' food intake and thereby affect the risk of developing malnutrition. These types of symptoms are categorised as nutrition impact symptoms [5]. Two questionnaires that have been designed to assess nutrition impact symptoms are the Disease Related Appetite Questionnaire (DRAQ) including questions about appetite, hunger, and other eating-related issues and the Eating Symptom Questionnaire (ESQ) including questions about symptoms such as nausea, pain, symptoms related to the gastrointestinal tract, swallowing difficulties, and changes in taste or smell. These questionnaires have been evaluated in patients with oncological and pulmonary

diseases [6,7]. They have also been applied in relation to patients with liver disease [8]. Based on the results from these studies, the questionnaires indicated that nutrition impact symptoms were common and associated with malnutrition.

Although one may expect that nutrition impact symptoms are the same in patients with burn injury as in other patient groups, it is rather the case that patients with burns are unique in epitomising the most severe model of trauma. Thus, other nutrition impact symptoms are likely to be present. Unfortunately, no instrument has been developed for use in patients after burn injuries that can capture all the information needed to assess nutrition impact symptoms. We suspected that there would be symptoms missing, not relevant, and/or that would need to be adjusted based on the original DRAQ and ESQ questionnaires for them to be suitable for these patients. The primary goal of this study was to develop and validate the two questionnaires for patients after burn injury.

## 2. Materials and Methods

This methodological study was intended to develop and validate DRAQ and ESQ for patients after burn injury. The design of the study is presented in Figure 1 and is based on the Boparai 2018 [9] guide on how to design and validate questionnaires.

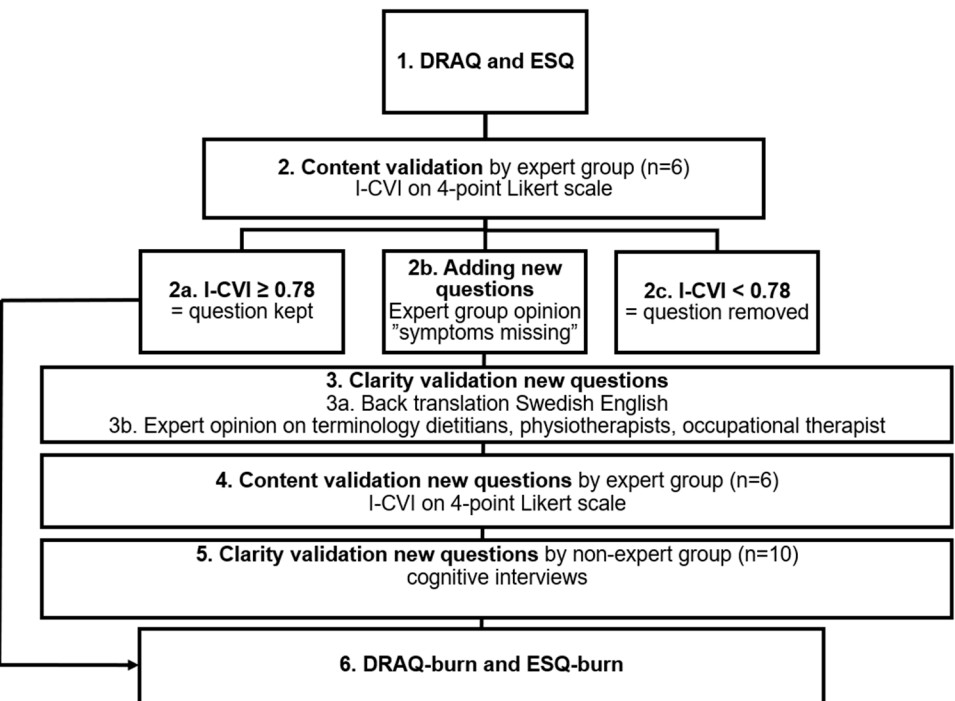

**Figure 1.** Flowchart of study design. Content and clarity validation process (1–6) of questionnaires Disease Related Appetite Questionnaire (DRAQ) and Eating Symptom Questionnaire (ESQ). Process described from original questionnaires DRAQ and ESQ (1) to modified questionnaires for patients after burn injury, DRAQ-burn and ESQ-burn (6). Content validation by expert group (2, 4) using content validity index (I-CVI) on 4-point Likert scale.

### 2.1. DRAQ and ESQ

The original DRAQ questionnaire includes 11 questions about the present status of appetite, hunger, and eating-related issues with fixed answers on a five-level scale. The ESQ lists 14 questions about symptoms that have been experienced during the previous two weeks related to eating and/or appetite. ESQ contains fixed answers on a five-grade scale from "no symptoms" to "severe symptoms" for thirteen of the questions and one open-ended question about any other symptoms that could affect appetite or prevent a person from eating.

English versions of DRAQ and ESQ were used in the current validation study. These versions were obtained from the translated Swedish version [6]. After careful rereading by two of the authors (AA and JD), minor adjustments were made for clarity. In ESQ, question 11, "no taste or strange taste" was added after the symptom description "change in taste". In DRAQ, for questions 6 and 11, the terms "healthy" and "disease" were changed to "before injury" and "injury" to match patients after burn injury.

### 2.2. Content Validation by Expert Group

The DRAQ and ESQ questionnaires were sent for content validity to twelve dietitians who had been chosen for their experience in burn care (expert group), in six different European countries. They were informed of the two questionnaires, the content validity process, and they were asked to answer the following questions:

(1) Are symptoms presented in DRAQ and ESQ representative for patients after burn injury?
(2) Are there any symptoms missing that should be added to DRAQ and ESQ?
(3) How relevant is each symptom for patients after burn injury on a 4-point Likert scale where 1 corresponds to not relevant and 4 to highly relevant?

Of the 12 dietitians, six answered the content validity questions and constituted the expert group. The expert group had worked an average of 14.5 years as dietitians, (min five years, max > 30 years) and in burn care for a mean of seven years, (min nine months, max 15 years). All of them worked in burn care facilities with both intensive care units and hospital wards specialised in treating patients with both minor and major burns. With one exception, all the experts working in these burn care facilities had the capacity to follow and advise their burn patients after discharge (5/6, 83%).

A content validity index (I-CVI) was calculated at the item level [10]. Each question (item) was evaluated by rating its relevance in Likert points (1 = not relevant, 2 = partially relevant, 3 = relevant, 4 = highly relevant). The individual expert evaluation of relevance Likert points, for each item, were dichotomised as: 1–2 Likert points = 0, 3–4 Likert points = 1. For each item, the I-CVI was computed using the dichotomised Likert points. An I-CVI ≥ 0.78 was considered to represent approved content validity and translated to ≥78% of the experts rated the relevance to be 3–4 (relevant or highly relevant) for that item [9]. An I-CVI ≥ 0.78 is a common cut-off for I-CVI [9,10]. Questions with I-CVI ≥ 0.78 were kept and questions with I-CVI < 0.78 were removed from the questionnaires. Symptoms missing in DRAQ or ESQ were added to the questionnaires and further analysed for content and clarity. New questions were suggested (Figure 1).

### 2.3. Clarity Validation of New Questions

To validate clarity, a back-translation from Swedish to English was made for the new questions added by the expert group. Expert opinions on terminology (from dietitians, physiotherapists, and an occupational therapist) were collected (*n* = 5).

### 2.4. Content Validation of New Questions by Expert Group

Symptoms that were added to a questionnaire were then tested for content validity by the expert group of dietitians (*n* = 6). They were asked to rate how relevant each new question was for patients with burns. The same 1–4 Likert scale dichotomisation of answers was used for the calculation of I-CVI. I-CVI ≥ 0.78 was considered to represent approved validity [9].

### 2.5. Clarity Validation of New Questions by Non-Expert Group

The new questions were also tested for clarity in a non-expert group through individual interviews with ten patients admitted to our hospital after burn injuries. They were asked 6–12 months after trauma how they interpreted the new questions [11,12].

*2.6. DRAQ-Burn and ESQ-Burn*

The modified questionnaires for patients after burn injuries, DRAQ-burn, and ESQ-burn were created by an expert group using questions with I-CVI ≥ 0.78 in content validation. The expert group approved the content and clarity validated new questions.

*2.7. Statistics*

A scale-content validity index average (S-CVI/Ave) was calculated for original content validation of DRAQ and ESQ as well as for the content validation of the newly added questions in DRAQ-burn and ESQ-burn. S-CVI is the average I-CVI value for all items on the scale, or, in other words, the average proportion of items rated as 3 or 4 by the experts [13]. All S-CVI were calculated on I-CVI and not on adjusted I-CVI. Davis [14] recommends 80% or better agreement among reviewers as a cut-off for new or revised instruments. S-CVI/Ave ≥ 0.8 was therefore considered the average congruity for DRAQ-burn and ESQ-burn [14].

## 3. Results

*3.1. Content Validity for DRAQ*

In DRAQ, nine questions had an original I-CVI ≥ 0.78 and were approved for content validity. Two of eleven questions (18%), questions 5 and 10, were not considered to be relevant for patients after burn injury (I-CVI < 0.78) and were removed from the questionnaire (Table 1).

**Table 1.** Content validity for DRAQ.

| Experts A-F (*n* = 6) Items/Questions | A | B | C | D | E | F | Number in Agreement | I-CVI * |
|---|---|---|---|---|---|---|---|---|
| 1. My appetite is . . . Answer: Very poor–very good | 1 | 1 | 1 | 1 | 1 | 1 | 6 | 1 |
| 2. My appetite varies from day to day–some days it's good, other days poor Answer: Fully agree–fully disagree | 1 | 1 | 1 | 1 | 1 | 1 | 6 | 1 |
| 3. When I eat . . . Answer: I feel full after eating a couple of bites of the meal–I usually don´t feel full | 1 | 1 | 1 | 1 | 1 | 1 | 6 | 1 |
| 4. I feel hungry . . . Answer: Never–always | 1 | 1 | 1 | 0 | 1 | 1 | 5 | 0.83 |
| 5. Food tastes . . . Answer: Very bad–very good | 1 | 0 | 1 | 0 | 0 | 0 | 2 | **0.33** |
| 6. Compared to before injury food tastes . . . Answer: A lot worse–a lot better | 1 | 0 | 1 | 1 | 1 | 1 | 5 | 0.83 |
| 7. I feel nauseated when I eat . . . Answer: Always–never | 1 | 1 | 1 | 1 | 1 | 1 | 6 | 1 |
| 8. How often do you eat anything? Answer: 0–2 times a day–more than 8 times a day | 1 | 1 | 1 | 1 | 0 | 1 | 5 | 0.83 |
| 9. My eating varies from day to day–some days I eat more, other days less). Answer: Fully agree–fully disagree | 1 | 1 | 1 | 1 | 1 | 1 | 6 | 1 |
| **10. Mostly I feel . . .** Answer: Very downhearted–very cheerful | 0 | 1 | 1 | 0 | 1 | 0 | 3 | **0.5** |
| **11. For how long has your injury affected your appetite?** Answer: Not at all–more than 6 months | 1 | 1 | 1 | 1 | 0 | 1 | 5 | 0.83 |

**Table 1.** *Cont.*

| Experts A-F (*n* = 6) Items/Questions | A | B | C | D | E | F | Number in Agreement | I-CVI * |
|---|---|---|---|---|---|---|---|---|
| **S-CVI/Ave DRAQ ** | | | | | 0.83 | | | |

\* Items rated on a four-point relevance Likert scale, answers dichotomised to 0 for response 1 or 2; 1 for response 3 or 4. Content is valid if ≥78% (Item content validity index, I-CVI ≥ 0.78) of experts think that the content is relevant. \*\* Scale content validity index/average, S-CVI/Ave, the combined number of items rated as "relevant" or "highly relevant" by all experts divided by the total number of ratings (55/66).

Five new questions were added to the modified DRAQ (DRAQ-burn) originating from comments on what was missing in the original DRAQ for patients after burn injury (Table 2).

**Table 2.** Content validity for suggested new questions to add to the DRAQ.

| Experts A-F (*n* = 6) Items/Questions | A | B | C | D | E | F | Number in Agreement | I-CVI * | Adjusted I-CVI ** |
|---|---|---|---|---|---|---|---|---|---|
| 1. Because of my burn injury I depend on someone else to prepare my meals | 1 | 0 | 1 | 1 | 1 | 1 | 5 | 0.83 | 1 |
| 2. I use oral nutrition supplements (prescribed from health-care provider) | 1 | 0 | 1 | 1 | 1 | 0 | 4 | **0.67** | 1 |
| 3. My dressings, scars, or itching affect(s) my appetite | 1 | 0 | 1 | 1 | 1 | 0 | 4 | **0.67** | 0.83 |
| 4. My dressings, scars, or itching prevent(s) me from eating | 1 | 0 | 1 | 1 | 1 | 0 | 4 | **0.67** | 0.83 |
| 5. My reduced functional ability prevents me from eating | 1 | 0 | 1 | 1 | 1 | 1 | 5 | 0.83 | 1 |
| S-CVI/Ave for new questions DRAQ *** | | | | | 0.73 | | | | |

\* Items rated on a four-point relevance Likert scale, answers dichotomised to 0 for response 1 or 2; 1 for response 3 or 4. Content is valid if ≥78% (Item content validity index, I-CVI ≥ 0.78) of experts think that the content is relevant. \*\* Considering "partially relevant" as an answer representative for patients after burn injury. Items rated on four-point relevance Likert scale, answers recalculated as, 0 for response 1; 1 for response 2, 3, or 4. \*\*\* Scale content validity index/average, S-CVI/Ave, the combined number of items rated as "relevant" or "highly relevant" by all experts divided by the total number of ratings (22/30). S-CVI/Ave calculated from I-CVI and not adjusted I-CVI.

The expert group validated 40% (2/5) of the new questions in DRAQ-burn as relevant for patients after burns I-CVI ≥ 0.78 (Table 2). Three questions had an I-CVI < 0.78 due to ratings < 3 by 2/6 experts (new question 2, 3, and 4 in Table 2). These three questions were kept in DRAQ-burn since out of six ratings <3, five were rated as partially relevant and only one was rated as not relevant (new question 4). Additionally, one expert (expert B, Table 2) self-reported a shorter time working in burn care, mostly with patients with minor burns (TBSA < 20%) and due to that "may be inclined to underestimate long-term symptoms". When recalculating I-CVI considering "partially relevant" as a response that should be representative for patients after burn injury, Likert point 1 = 0 and Likert point 2, 3, or 4 = 1, I-CVI was ≥0.78 for all new questions in DRAQ-burn (see "adjusted I-CVI", Table 2).

*3.2. Content Validity for ESQ*

In the original ESQ, five questions had an I-CVI ≥ 0.78 and were approved for content validity. Eight out of fourteen questions (57%), questions 2–7 and 11–12, were considered to not be relevant for patients after burn injury (I-CVI < 0.78) and were removed from the questionnaire (Table 3).

Two new questions were added to the modified ESQ (ESQ-burn) originating from comments on what was missing in the original ESQ for patients after burn injury (Table 4).

**Table 3.** Content validity for ESQ.

| Experts A-F (*n* = 6) Items/Questions 1–13. During the Last 2 Weeks Have You Had the Following Symptoms? [6] | A | B | C | D | E | F | Number in Agreement | I-CVI * |
|---|---|---|---|---|---|---|---|---|
| 1. Nausea | 1 | 1 | 1 | 1 | 1 | 0 | 5 | 0.83 |
| 2. Vomiting | 0 | 1 | 1 | 1 | 1 | 0 | 4 | **0.67** |
| 3. Stomach ache | 0 | 0 | 1 | 0 | 0 | 1 | 2 | **0.33** |
| 4. Diarrhoea | 0 | 1 | 1 | 0 | 1 | 1 | 4 | **0.67** |
| 5. Constipation | 0 | 1 | 1 | 0 | 1 | 1 | 4 | **0.67** |
| 6. Pain in the mouth | 1 | 0 | 1 | 1 | 1 | 0 | 4 | **0.67** |
| 7. Dry mouth | 0 | 1 | 1 | 1 | 0 | 0 | 3 | **0.5** |
| 8. Pain or ache affecting my appetite | 1 | 1 | 1 | 0 | 1 | 1 | 5 | 0.83 |
| 9. Difficulties chewing | 1 | 1 | 1 | 1 | 1 | 0 | 5 | 0.83 |
| 10. Difficulties swallowing | 1 | 1 | 1 | 1 | 1 | 0 | 5 | 0.83 |
| 11. Changes in taste | 1 | 0 | 1 | 0 | 1 | 1 | 4 | **0.67** |
| 12. Affected by smell | 1 | 0 | 1 | 0 | 1 | 1 | 4 | **0.67** |
| 13. Pain or ache preventing me from eating | 1 | 1 | 1 | 0 | 1 | 1 | 5 | 0.83 |
| S-CVI/Ave ESQ ** | | | | 0.69 | | | | |

\* Items rated on a four-point relevance Likert scale, answers dichotomised to 0 for response 1 or 2; 1 for response 3 or 4. Content is valid if ≥78% (Item content validity index, I-CVI ≥ 0.78) of experts think that the content is relevant. ** Scale content validity index, S-CVI/Ave, the combined number of items rated as "relevant" or "highly relevant" by all experts divided by the total number of ratings (54/78).

**Table 4.** Content validity for suggested new questions to add in the ESQ.

| Experts A-F (*n* = 6) Items/Questions | A | B | C | D | E | F | Number in Agreement | Item ICV * |
|---|---|---|---|---|---|---|---|---|
| 1. Tiredness, low energy (fatigue) affecting my appetite | 1 | 0 | 1 | 1 | 1 | 1 | 5 | 0.83 |
| 2. Tiredness, low energy (fatigue) preventing me from eating | 1 | 0 | 1 | 1 | 1 | 1 | 5 | 0.83 |
| S-CVI/Ave for new questions ESQ ** | | | | 0.83 | | | | |

\* Items rated on a four-point relevance Likert scale, answers dichotomised to 0 for response 1 or 2; 1 for response 3 or 4. Content is validated if ≥78% (Item content validity index, I-CVI ≥ 0.78) of experts think that the content is relevant. ** Scale content validity index/average, S-CVI/Ave, the combined number of items rated as "relevant" or "highly relevant" by all experts divided by the total number of ratings (10/12).

The expert group validated the new questions in ESQ-burn as relevant for patients after burn injury I-CVI ≥ 0.78 (Table 4).

### 3.3. Clarity Validity for New Questions in DRAQ and ESQ

Minor adjustments to terminology were made after back-translation and expert opinions by health care professionals (*n* = 5).

Ten patients (non-expert group) answered and explained how they interpreted the new questions in DRAQ-burn and ESQ-burn. Minor adjustments to clarify questions were made after the interviews, for example, underlined phrases: "prevents me from eating" and "affecting my appetite".

### 3.4. DRAQ-Burn and ESQ-Burn

The modified questionnaires for patients in the rehabilitation phase after burn injury consisted in the final versions of 14 questions (Supplementary Materials File S1) and eight questions (Supplementary Materials File S2). ESQ-burn had seven closed-ended questions and one open-ended question, where patients could add any other symptoms that could

affect their appetite or prevent them from eating. A high degree of consensus that DRAQ-burn and ESQ-burn are relevant for assessing nutrition impact symptoms post burn was reached in the expert group, where S-CVI/Ave for DRAQ-burn = 0.86 (72/84) and for ESQ-burn of 0.83 (35/42).

## 4. Discussion

Based on our content validation study, we suggest DRAQ-burn and ESQ-burn as two new questionnaires specifically adapted to explore the prevalence of nutrition impact symptoms in patients after burn injury.

Nutrition impact symptoms and specifically appetite can be measured through several different questionnaires: The Appetite Hunger and Sensory Perception questionnaire (AHSP), Council on Nutrition Appetite Questionnaire (CNAQ) and its short version SNAQ (the Simplified Nutritional Appetite Questionnaire) [15,16]. There are also several checklists used to evaluate nutrition impact symptoms, developed for specific medical conditions [17–20]. However, neither of these have been designed nor validated to investigate nutrition impact symptoms post-burn injury. DRAQ and ESQ are derived from CNAQ. The reason why these questionnaires were selected in this study was because they were already translated into Swedish. Additionally, they have been used to evaluate appetite and nutrition impact symptoms in several other medical conditions and studies [6,8,21].

The expert group, all dietitians with experience in burn care, were chosen due to the nature of the questionnaires. Evaluating nutrition impact symptom prevalence and severity is a part of the nutritional assessment described in the nutritional care process used by dietitians [22]. Initially, twelve dietitians were asked to take part in the expert group. Of those twelve, only six answered the content validation protocol. This may be a limitation of the study. However, an expert group consisting of 3–10 experts is considered to be sufficient for this form of validation [9]. The content validation was written in English and the clarity validation in Swedish. We used this approach due to the limited number of dietitians with burn care experience working in Sweden (*n* = 2), which ruled out an expert group of Swedish-only dietitians. To close the gap, a back-translation from English—Swedish was undertaken. A clarity validation study should be performed in a native-English speaking non-expert group before using these questionnaires in English.

By using an expert group, several questions in the two original questionnaires were found to be irrelevant for the burn group. Other symptoms more specific to the issues of burn patients were missing. The expert group suggested that some questions be removed and other more relevant (to burns) questions such as fatigue and functional ability be added. Although developed to be generic and able to capture nutrition impact symptoms for an array of medical conditions, DRAQ and ESQ were specifically designed to capture symptoms in patients with the risk of malnutrition related mostly to pulmonary disease and cancer [6]. Therefore, many of the original questions in DRAQ (2/11, 18%) and ESQ (8/14, 57%) had to be changed. Symptoms in ESQ such as vomiting, diarrhoea, constipation, dry mouth, or changes in taste are well known symptoms in oncological and oncological treatment-related conditions. These symptoms are not common in patients with burns. One could argue, though, that symptoms in the original ESQ are probably more commonly present in the acute phases after trauma and rarer in the rehabilitation phase.

When calculating I-CVI in DRAQ-burn and ESQ-burn, only symptoms considered relevant or highly relevant by the expert group were calculated as 1 in the equation. Not calculating I-CVI from original symptoms rated as "partially relevant" as 1 led to the removal of several symptoms in DRAQ and ESQ and could potentially mean that these nutrition impact symptoms are missed for some individuals during follow-up after burn injury. New questions considered relevant for patients after burn injury were added. The final version of DRAQ-burn and ESQ-burn consisted of 14 and eight questions, respectively, all considered relevant to patients after burn injury. When analysing DRAQ content validity, only 2/5 suggested new questions (I-CVI ≥ 0.78). These were considered relevant according to our rating system. The remaining three new questions were kept in DRAQ-burn although

the I-CVI < 0.78. This was due to several reasons in the analysis. New questions considered "partially relevant" came from the expert group. One expert self-reported a shorter time working in burn care, which was mostly with patients with minor burns (TBSA < 20%). This expert recognised that she "may be inclined to underestimate long-term symptoms". This expert's observation resulted in keeping these three questions in DRAQ-burn. To our knowledge, there are no studies on nutritional impact symptoms post-burn injury. One can argue though, for the addition of the new questions to DRAQ-burn and ESQ-burn considering that there have been reports in other patient groups on a relationship between reduced functional capacity, reduced handgrip strength and the prevalence of nutritional impact symptoms [5,7]. Although keeping three questions with I-CVI < 0.78 in DRAQ, the S-CVI/Ave for DRAQ-burn (0.86) showed a good content validity for the questionnaire as well as the S-CVI/Ave for ESQ-burn (0.83) [14]. I-CVI for each item greater than ≥0.78 and S-CVI/Ave ≥ 0.9 is needed for the scale to be regarded as having an excellent validity [23]. Since the clarity validation by the non-expert group only resulted in minor adjustments to the questionnaires, new questions were regarded as well-defined. During the interviews, patients talked about the questions with their own burn injury in mind. From the discussions, it became clear that it is always good to have a health professional available to answer any questions about the questionnaires and phrasings therein.

A future step is to use DRAQ-burn and ESQ-burn to explore the prevalence of nutrition impact symptoms post-burn injury, the association with dietary intake and risk of malnutrition and its role within nutritional care in the rehabilitation phase of post-burn injury. It would be useful in future studies and also in regular follow-up visits in the rehabilitation phase post-burn injury to use DRAQ-burn and ESQ-burn in order to optimise nutritional intake in an effort to improve the outcome after burn injuries. This would increase our ability to identify and possibly treat symptoms that can negatively affect the patients' food intake and thereby affect the risk of developing malnutrition.

## 5. Conclusions

DRAQ and ESQ were modified and adapted for patients in the rehabilitation phase after burn injury. Eighteen and 57% of the questions in DRAQ and ESQ, respectively, were considered irrelevant and thus removed and more relevant questions added. Within the expert group, a high degree of consensus related to DRAQ-burn and ESQ-burn was reached. The results indicate that modified questionnaires can be useful for assessing nutrition impact symptoms in patients after burn injury.

**Supplementary Materials:** The following supporting information can be downloaded at: https://www.mdpi.com/article/10.3390/ebj3010013/s1, Supplementary Materials File S1: DRAQ-burn; Supplementary Materials File S2: ESQ-burn.

**Author Contributions:** Conceptualisation, methodology, software, validation, and formal analysis, resources, data curation, A.A., A.M., F.H. and J.D.; Investigation, J.D.; Writing—original draft preparation, writing—review and editing, A.A., A.M., F.H. and J.D.; Visualisation, A.M. and J.D.; Supervision, A.A., A.M. and F.H.; Project administration, F.H. and J.D.; Funding acquisition, F.H. All authors have read and agreed to the published version of the manuscript.

**Funding:** This research received no external funding from agencies in the public, commercial, or not-for-profit sectors. However, support from Uppsala County Council (ALF) was received.

**Institutional Review Board Statement:** The study was conducted according to the guidelines of the Declaration of Helsinki and approved by the Regional Ethical Review Board in Uppsala Dnr 2018/436 (approved 11 November 2018) and Dnr 2020-02088 (Approved 3 June 2020).

**Informed Consent Statement:** Not applicable. Ten patients were, during clinical routine assessment by the dietitian, asked how they interpreted the new questions, and only oral informed consent was obtained for this.

**Data Availability Statement:** Data are contained within the article.

**Acknowledgments:** We wish to acknowledge the help provided by the expert and non-expert groups. We also acknowledge the help provided in designing the study by the staff and doctoral students at Department of Food Studies, Nutrition and Dietetics, Uppsala University, Sweden. We thank Frode Slinde, Heléne Bertéus Forslund, and Ola Wallengren for answering questions about the original DRAQ and ESQ and allowing us to modify the questionnaires for patients after burn injury. We also thank our colleagues at Uppsala University Hospital for their valuable help with the terminology and Lena Martin for her valuable support.

**Conflicts of Interest:** The authors declare no conflict of interest.

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
