# Peer review of "Two Modified Questionnaires for the Assessment of Nutrition Impact Symptoms in the Rehabilitation Phase after Burn Injury: A Content Validation Study"

_2673-1991, doi:10.3390/ebj3010013_

Round 1

Reviewer 1 Report

Please have this manuscript reviewed by a native English speaker prior to resubmission as there are numerous grammar and syntax errors throughout the paper.

In the introduction, the authors state that the primary goal was to validate the DRAQ and ESQ but did not expect these to be suitable for burn patients. However, there is no mention of modifying the existing DRAQ and ESQ to make these scales more appropriate for the burn population. Please add this to the study objectives, since that is the main focus of the paper.

I believe the authors' methodology may be flawed. The explanation of the methods is not clear and is confounded by the English language errors. Please rewrite the methods section more clearly. Please provide explanation for the I-CVI and S-CVI cutoffs that were chosen. For the content validation aspect, the manuscripts states that symptoms were added to the questionnaire based on expert opinion. Is there literature available to further support the addition of these symptoms?

In the discussion section, please comment on the expected and/or recommend clinical use of these questionnaires.

Author Response

We would like to thank you for valuable and appreciated input on our manuscript.

Reviewer #1: Please have this manuscript reviewed by a native English speaker prior to resubmission as there are numerous grammar and syntax errors throughout the paper.

Answer: We thank reviewer #1 for valuable input. Manuscript has now been reviewed by a professional translator, Andrea Smith, and changed accordingly.

Reviewer #1: In the introduction, the authors state that the primary goal was to validate the DRAQ and ESQ but did not expect these to be suitable for burn patients. However, there is no mention of modifying the existing DRAQ and ESQ to make these scales more appropriate for the burn population. Please add this to the study objectives, since that is the main focus of the paper.

Answer: Good point, we have now changed in our objective to” develop and validate DRAQ and ESQ” to clarify this (page 1, line 20 and page 2 line 70).

Reviewer #1: I believe the authors' methodology may be flawed. The explanation of the methods is not clear and is confounded by the English language errors. Please rewrite the methods section more clearly. Please provide explanation for the I-CVI and S-CVI cutoffs that were chosen.

Answer: We have corrected grammar and syntax errors in the method section. We have also revised the manuscript and added more to explain I-CVI and S-CVI cut-offs to make it clearer (page 4, line 123 and line 154-156).

Cut-off for I-CVI ≥ 0.8 is recommended as a minimum by Davis (1992, reference 14). Lynn´s (1986) criteria with a minimum of 0.78 for 6 to 10 experts is also what Boparai 2018 (reference 9) recommends in their guide on how to design and validate questionnaires. We chose I-CVI ≥ 0.78 to represent approved validity which we describe in the method section. For the original DRAQ and ESQ questions with I-CVI ≥ 0.78 were kept and questions with I-CVI < 0.78 were removed from the questionnaires. All questions kept in DRAQ-burn and ESQ-burn had I-CVI 0.83-1.0.

Of the five new suggested questions to DRAQ-burn, two had I-CVI 0.83 but three questions had I-CVI of 0.67. Two of the six experts gave ratings < 3 on these three new questions and should according to cut-offs be removed from the questionnaires. One are though advised to analyze the ratings, and when doing so, we saw that of six possible ratings < 3, five were rated as partially relevant and only one rated as not relevant. One of these two experts self-reported a shorter time working in burn care mostly with patients with minor burns (TBSA < 20 %) and due to that “may be inclined to underestimate long term symptoms”. Due to those observations, we recalculate I-CVI considering “partially relevant” as a response that could be representative for patients after burn injury, I-CVI was then ≥ 0.78 (0.83-1) for all new questions in DRAQ-burn. And although we kept these three new questions in DRAQ-burn S-CVI/Ave was high 0.86 (calculated on I-CVI and not on adjusted I-CVI) which indicated a good content validity for DRAQ-burn.

 Both new suggested questions to ESQ-burn had I-CVI 0.83. An S-CVI/Ave for ESQ-burn were 0.83 indicating a good content validity for ESQ-burn.

Concerning the cutoff for S-CVI/Ave, we have chosen ≥ 0.8 which is often considered average congruity for this index (Davis 1992, reference 14). As we discuss in the discussion section our scale does not fulfill the criterium to be considered having an excellent validity S-CVI/Ave >0.9 (Polit 2006, reference 13, Polit 2007, reference 23).

Reviewer #1: For the content validation aspect, the manuscripts states that symptoms were added to the questionnaire based on expert opinion. Is there literature available to further support the addition of these symptoms?

Answer: Thank you for a valid question. New questions added covers areas such as depending on someone else to prepare one’s meals, using oral nutritional supplements, dressing, scars, itching affecting appetite or preventing patients from eating or functional ability preventing from eating as well as tiredness affecting appetite or preventing from eating.

There is to our knowledge no studies on nutritional impact symptoms or how they affect nutritional status post-burn injury. And therefore, neither no literature in the burn field to support the added questions. Thereof the importance of developed instrument to measure nutritional impact symptoms post burn-injury. We can´t find any similar studies post other intensive care treatment or traumas either. But there are observations from other areas such as patients with head and neck cancer (Kubrak 2010, reference 5) which indicate that having nutritional impacts symptoms before treatment may adversely affect both dietary intake, weight as well as functional capacity of patients. Another example is patients with neuroendocrine tumors were 25 % had impaired hand grip strength which were associated with specific nutritional impact symptoms (Borre 2018, reference 7). We have added this to the discussion section (page 10, line 300-303).

Reviewer #1: In the discussion section, please comment on the expected and/or recommend clinical use of these questionnaires.

Answer: In the discussion section we have clarified our comment on the expected/recommended clinical use of these questionnaires (page 10, line 317-320).

Reviewer 2 Report

Thank you for the opportunity to review this manuscript – I found the topic to be very important, and the methods applied to be appropriate. It is a pleasure to recommend this excellent paper for publication.

This article aims to adapt (or perhaps more accurately develop) the Disease Related Appetite Questionnaire (DRAQ) and Eating Symptom Questionnaire (ESQ) for use in patients recovering from burn injury.

The introduction is well written, concise and comprehensive, with a clear through line as to why nutrition is important in burn recovery, and why it would be important to be able to track nutrition impact symptoms. It is also clear as to why burn patients may require an adapted version that has been developed for use specifically in this population.

The methods appear to me to be both interesting and rigorous, and I have every confidence that subsequent patient-based validation will bear out the expert-based content validity established by the process and data presented in this manuscript.

As with the introduction, the discussion is clearly written and both concise and comprehensive. This is a very well-written manuscript that does an excellent job of covering an important topic – I congratulate the authors on an excellent paper. I think these tools – having been adapted and developed for use in burns – hold the potential to make an impact both in clinical care as well as in research by increasing our ability to identify, intervene, and track nutritional challenges facing patients as they recover.

My only suggestion is that further psychometric testing be performed to ensure reliability and validity when these measures are deployed in actual patient populations.

Author Response

We would like to thank you for valuable and appreciated input on our manuscript.

Thank you for good remarks, we have changed ”adapt” to ”develop” DRAQ and ESQ in our aim (page 1, line 20 and page 2, line 70). We will look into the psychometric testing going further.

Reviewer 3 Report

The authors have analyzed two validated questionnaires on symptoms that affect nutritionand specially modified for the evaluation of burn patients. Nutrition plays a crucial role both in the acute phase and in the rehabilitation phase of burn patients. In order to be able to carry out a scientific evaluation of specific symptoms that influence nutrition, validated questionnaires are essential.

In the present work, the adaptation of the questionnaires was carried out in a methodologically flawless manner. In my opinion, the number of surveyed experts and non-experts could have been higher. Nevertheless, the results represent a useful catalog of Questions for the evaluation of burn patients.

Author Response

We thank reviewer#3 for good comments.

The small number of experts may be considered as a limitation as we also do bring up in the discussion (page 9, line 260-263). However, the number of six experts are in the interval of the recommended 3–10 experts considered enough for this form of validation. In the non-expert group (n=10) they understood the questions as they were intended, and only minor adjustments were made like underlining the phrase” affect my appetite” in the questionnaire. Since no more new comments were made, we stopped our interviews after including ten in the non-expert group.

Reviewer 4 Report

The authors presented modified nutrition impact assessment methodology in burn patients. The work is significant to understand the patient rehabilitation and nutrition impact. Did the authors correlate burn severity to the nutrition in any of their assessments which may impact rehabilitation outcome?  

Author Response

We thank reviewer #4 for a valid question, we did not correlate burn severity to the nutrition. We are now going to continue our research and use DRAQ-burn and ESQ-burn in the rehabilitation phase post burn-injury to investigate prevalence of nutritional impact symptoms. Then it would be interesting to correlate burn severity to prevalence of nutritional impact symptoms.

Round 2

Reviewer 1 Report

I thank the authors for addressing my previous comments. At this time, I have no additional comments on this manuscript.